# Integrated GNSS/IMU-Gyrocompass with Rotating IMU. Development and Test Results

**Gennadiy Emel'yantsev** [1,2], **Oleg Stepanov** [1,2,*], **Aleksey Stepanov** [1,2], **Boris Blazhnov** [2], **Elena Dranitsyna** [1,2], **Mikhail Evstifeev** [1,2], **Daniil Eliseev** [1,2] **and Denis Volynskiy** [2]

[1] International Laboratory "Integrated Navigation and Attitude Reference Systems", ITMO University, 49 Kronverkskiy pr., 197101 St. Petersburg, Russia; emeliantsev_gi@mail.ru (G.E.); apstepanov@itmo.ru (A.S.); evdranitsyna@itmo.ru (E.D.); mievstifeev@itmo.ru (M.E.); dpeliseev@itmo.ru (D.E.)

[2] Concern CSRI Elektropribor, JSC, 30 Malaya Posadskaya St, 197046 St. Petersburg, Russia; blazhnov_b@mail.ru (B.B.); silent_d@mail.ru (D.V.)

[*] Correspondence: soalax@mail.ru

**Abstract:** The paper presents the developed integrated GNSS/IMU gyrocompass which, unlike the existing systems, contains a single-axis rotating platform with two antennas installed on it and an inertial measurement unit with tactical grade fiber-optic gyros. It is shown that the proposed design provides attitude solution by observing the signals of only one navigation satellite. The structure of the integrated GNSS/IMU gyrocompass, its specific features and prototype model used in the tests are described. The given test results in urban conditions confirmed heading determination accurate to ±1.5° (3σ).

**Keywords:** fiber-optic gyro; two-antennas tightly-coupled system; GNSS/IMU-gyrocompass; single difference of phase measurements; multi-GNSS (GPS/GLONASS); common clock; single-frequency receivers

---

## 1. Introduction

In a wide range of problems involving mapping, exploration of North areas of the World Ocean, active application of robotic technology, unmanned aerial vehicles and automobiles, and determination of the vehicles attitude is required [1–4]. Traditionally, data from inertial measurement units (IMU) and strapdown inertial navigation systems (SINS) are used for solution of this problem, which are completely autonomous but unfortunately have an error accumulating with time [5–7]. Recently, devices contains at least one baseline formed by two global navigation satellite systems (GNSS) antennas have been widely used for attitude determination of mobile objects. [8–11], the so-called GNSS compasses. These measurements contain data on the angle between the satellite direction and the vector formed by GNSS antennas [8,10–13]. GNSS compass advantages include no error accumulation, acceptable accuracy, and relatively low cost [14]. However, their accuracy and reliability are greatly degraded due to satellite signal outages, unfavourable observation geometry, and multipath propagation, which is inevitable particularly with highly dynamic object and in signal degraded conditions such as urban canyons [15–19].

To overcome these drawbacks, attitude determination systems integrating IMU and GNSS compass data are developed [6,7,20], we will call them as integrated GNSS/IMU gyrocompass or simply GNSS/IMU gyrocompass. IMU rotation is important for improving the performance of these integrated systems, because rotation provides observability of IMU errors and finally enhances the attitude accuracy [21–25].

One of major problems in implementation of GNSS/IMU compasses using phase measurements is ambiguity resolution, i.e., finding the component equal to the integer number of carrier wave lengths [8,9,11,26]. Applying two- and three-frequency receivers makes it possible to resolve the phase ambiguity [27,28]; however, it increases the system cost and power consumption. With a single-frequency receiver, to enhance the chance of phase ambiguity the redundant number of navigation satellites from different GNSS can be observed (multi-GNSS), including the Russian GLObal Navigation Satellite System (GLONASS), the Chinese BeiDou navigation satellite system (BDS), and the European Galileo navigation satellite system [9,15,27,29]. However, studies conducted by the authors [30] demonstrate that if the number of various GNSS satellites in the visibility zone is considerable, not all of them are effective for the heading determination. The most effective satellites are the ones with elevation of 15° to 75° and with azimuth within ±45° to the normal of antenna baseline (green areas in Figure 1). Note that the satellites are situated in the zone satisfying the above conditions for a short time because they move with respect to GNSS/IMU compass antenna module. Thus, even in favorable reception conditions the number of satellites will be limited due to ineffective zones, not to mention the urban canyons.

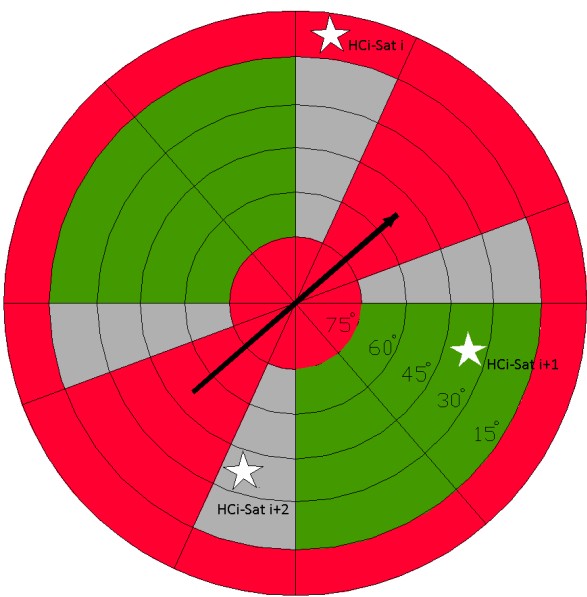

**Figure 1.** Diagram presents the effectiveness of the observed satellites to determine the direction of antenna baseline (in black). Effective zones are shown in green, low effective, in grey, and ineffective, in red.

Therefore, in designing integrated systems it is important to reduce the number of satellites sufficient for attitude solution. As shown in the papers including those written by the authors, rotation of satellite antennas—resulting in phase ambiguity resolution—helps to reduce the number of required satellites [31–33]. References [30,31] contain the research results for a compact (antenna baseline of 0.3 m) integrated GNSS/IMU compass with single-axis antennas and IMU rotation. Experimental data reveal heading accuracy better than 3° (3σ) and roll and pitch angles' accuracy, better than 0.3° (3σ). Note that the algorithms proposed in [32,33] use the double phase differences, whose generation requires simultaneous observation of min two satellites [17,18,32,34].

As opposed to the common designs, the proposed integrated GNSS/IMU gyrocompass provides heading solution by observing the signals from only one satellite due to the use of single-axis rotating platform with installed IMU and two satellite receiving antennas with two satellite navigation receivers. What is important, the receivers use a common clock, which reduces phase measurement errors when generating the single phase differences [35]. IMU rotation and application of tactical grade fiber-optic

gyros (FOGs) helps to keep the heading, roll, and pitch accuracy during long GNSS signal outages. Existing integrated GNSS/IMU compasses operate with the desired accuracy after signal outage for max units of minutes, which is insufficient in degraded GNSS conditions. It happens for example in urban conditions or hilly areas and when passing narrows or a port in sea navigation. In this case, it is desirable to ensure the required accuracy of the system after satellite signals outage for a time of about one hour.

Some development stages of the integrated GNSS/IMU gyrocompass have already been discussed in a number of previous publications; however, they covered only partial development aspects [35] and assumed that data from IMU on microelectromechanical system (MEMS) gyros are employed [31], and that tests are conducted on a test bench [30]. This research provides the most complete description of the features of the integrated GNSS/IMU gyrocompass and test results for its prototype model installed onboard an automobile in real urban conditions, which confirm the system operability and advantages.

The paper is structured as follows. Section 2 introduces the main notations and reference frames. Section 3 describes the structure of the integrated GNSS/IMU gyrocompass and its features, and the prototype model used for the tests. Error models of the applied sensors and the operation algorithm of the integrated GNSS/IMU gyrocompass are given in Section 4. Sections 5 and 6 presents and analyzes the experimental results. Section 7 summarizes the paper.

## 2. Reference Frame Definitions and Main Notations

Below we introduce the notations for the reference frames used in the paper (see Figure 2):

$X_iY_iZ_i$—inertial frame (i-frame);
$X_eY_eZ_e$—Earth-Centered-Earth-Fixed Frame (e-frame);
*ENU*—n-frame, a local geographic frame (eastward-northward-upward);
$X_bY_bZ_b$—b-frame bound with the integrated GNSS/IMU gyrocompass body;
$X_sY_sZ_s$—s-frame bound with the antenna baseline and measurement axes of accelerometers and gyros, with the antenna baseline $\vec{B}$ oriented along axis $Y_s$ (Figure 3), s-frame rotates with respect to b-frame about axis $Z_b$ coinciding with axis $Z_s$ at angular velocity $\dot{u}$.

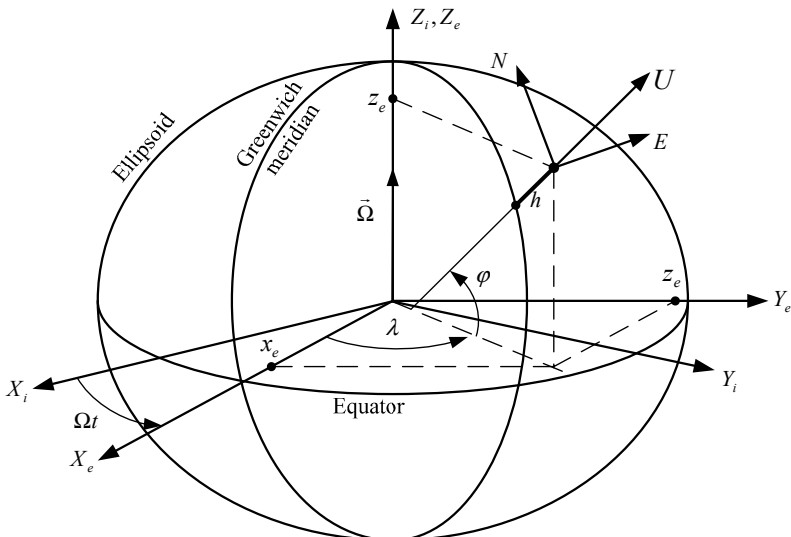

**Figure 2.** Reference frames: $X_iY_iZ_i$—inertial frame (i-frame), $X_eY_eZ_e$—Earth-Centered-Earth-Fixed Frame (e-frame), *ENU*—eastward-northward-upward (n-frame), $\Omega$—angular velocity of Earth about its polar axis, $t$—time.

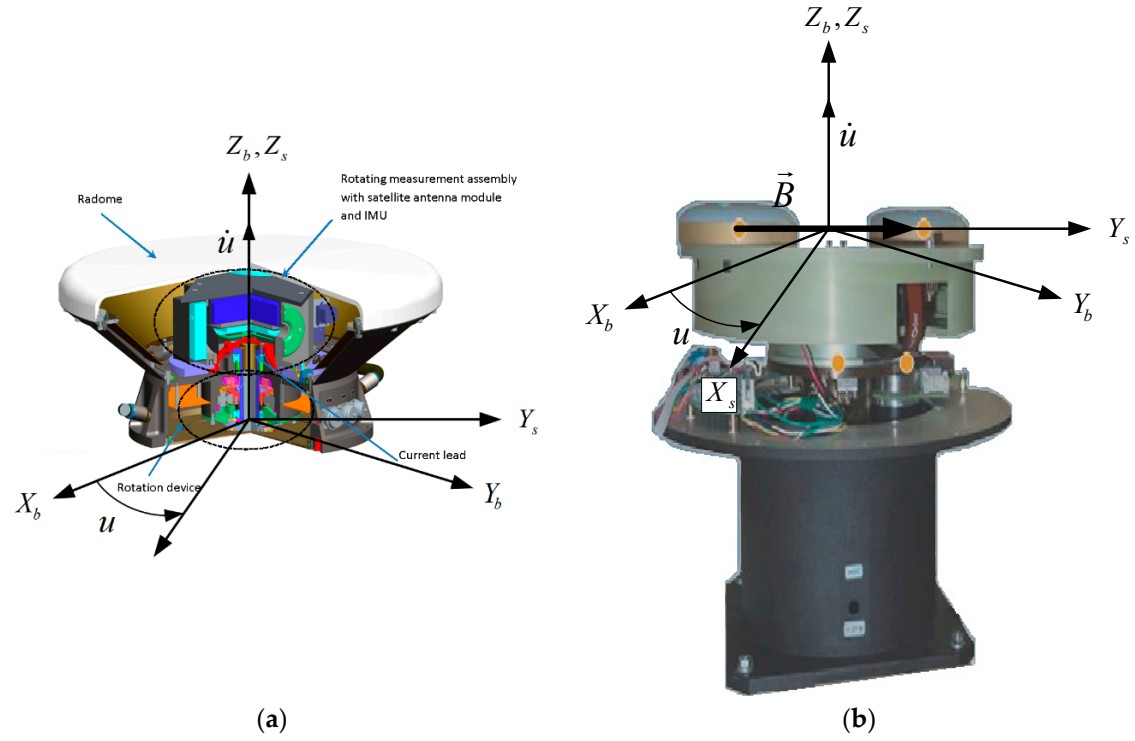

(a)                                              (b)

**Figure 3.** Appearance of integrated GNSS/IMU gyrocompass (**a**) and its prototype GNSS compass include rotating antenna unit (**b**), and reference frames bound with them: b-frame $X_bY_bZ_b$ and s-frame $X_sY_sZ_s$.

Also we assume that

$\mathbf{p} = (\phi, \lambda, h)^T$ and $\delta\mathbf{p} = (\delta\varphi, \delta\lambda, \delta h)^T$ are the vectors of geographic coordinates and their errors;

$\mathbf{V_n^e} = [V_E, V_N, V_U]^T$, $\delta\mathbf{V_n^e} = [\delta V_E, \delta V_N, \delta V_U]^T$ is the vector of object linear velocity with respect to the Earth projected onto n-frame axes and its error vector;

$K, \psi, \theta$ are the heading, roll and pitch angles;

$\beta, \gamma, \alpha$ are INS errors in modeling a local geographic frame; $\beta$ and $\gamma$ are pitch and roll errors, and $\alpha$ is a heading error;

$\boldsymbol{\omega_s^{i,s}} = [\omega_{X_s}, \omega_{Y_s}, \omega_{Z_s}]^T$ and $\delta\boldsymbol{\omega_s^{i,s}} = [\delta\omega_{X_s}, \delta\omega_{Y_s}, \delta\omega_{Z_s}]^T$ are the projections of gyros angular velocity with respect to i-frame projected onto their measurement axes s-frame and their errors;

$\mathbf{f_s} = [f_{X_s}, f_{Y_s}, f_{Z_s}]^T$ and $\delta\mathbf{f_s} = [\delta f_{X_s}, \delta f_{Y_s}, \delta f_{Z_s}]^T$ are the specific force vector projected onto its measurement axes (s-frame) and its errors;

$\rho^i, \dot{\rho}^i, \Delta\phi^i$ are the pseudorange, radial velocity and single difference of carrier phase received by the spaced GNSS antennas for each observed i-th satellite.

## 3. Materials

The developed GNSS/IMU gyrocompass is a tightly coupled integrated GNSS/INS system using satellite signals with both code and frequency division (GPS/GLONASS). Below we describe the gyrocompass structure, its specific features and the prototype used in the tests.

### 3.1. Integrated GNSS/IMU Gyrocompass Structure and Features

The integrated GNSS/IMU gyrocompass comprises two satellite antennas, IMU and a computing unit installed on a rotating foundation providing rotation about the vertical axis $Z_b$. Data can also be received from external sensors of the object linear velocity and altitude (depth).

The antenna module contains two antennas with two satellite navigation receivers receiving GNSS signals with code division (GPS) and frequency division (GLONASS) and using a common

clock. The clock frequency stability is better than $10^{-6}$. It should be emphasized that instead of MEMS gyros traditionally applied in existing GNSS compasses the IMU contains three more accurate small-sized tactical grade open loop FOGs (CJSC Fizoptika, Russia) [36] with drift instability of 6–30°/h. Bias instability of the employed accelerometers is estimated to be 0.001 g. 3D model of the developed integrated GNSS/IMU gyrocompass is shown in Figure 3a.

Major distinctive features of the developed integrated GNSS/IMU gyrocompass include rotating antenna unit and IMU installed on a common foundation, common clock, and attitude determination method using phase measurements from two antennas. Rotation of antenna baseline excludes rather complicated search algorithms to estimate phase ambiguity and improves heading accuracy because the observed satellites are located in elevation and azimuth zones effective for heading determination for a much longer time [9]. Due to rotation of antenna module the direction of antenna baseline with respect to satellite direction is changing, which critically reduces the error and provides heading solution by observing the signals of only one satellite.

IMU rotation creates the required dynamics and makes it possible to decompose the gyro error model and effectively estimate its components if GNSS data are available. Selection of the motion law is a separate problem not covered in the paper. It should be only noted that angular displacement is realized using an original optimal algorithm effective in terms of minimizing the errors of GNSS/IMU gyrocompass. Optimization of rotation parameters is detailed in [31].

These features (rotation and using IMU on tactical grade FOGs) allow downsizing the system, with its overall dimensions being Ø400 × 270 mm. The proposed distance between the antennas' phase centers is 0.36 m. The majority of existing GNSS/IMU compasses have larger dimensions and antenna baseline of 0.7–1.5 m, which is required to obtain the heading solution accurate to 0.5–1°(3$\sigma$) [37,38].

INS and GNSS data are integrated via extended Kalman filter (EKF) described in Section 4.4 using a tightly coupled scheme [18,35]. As raw data for joint processing we use pseudorange $\rho^i$, radial velocity $\dot{\rho}^i$, and single difference of carrier phase $\Delta\phi^i$ received by the spaced antennas for each observed i-th satellite. The available measurements are processed using so-called invariant scheme, i.e., based on complementary filter. Thus, in filtering described further, the measurement errors are estimated by using differential measurements [39,40]. The data sources used in this study are INS, satellite receivers, and external sensors of linear velocity and altitude (depth) in case of satellite signal outage.

Note that phase measurements are processed using their single differences, which allows getting heading updates by signals from only one satellite. Then to eliminate error in phase single differences arising due to various receiver clock errors of two antennas, two satellite navigation receivers with a common external clock are applied [35]. It should be pointed out that using the phase single difference requires no additional (special) processing of satellite signals with frequency division (GLONASS).

One should underline that with GNSS data available, the applied more accurate FOGs can be calibrated by rotation with the accuracy sufficient to guarantee the acceptable accuracy of INS attitude with completely lost satellite data for a rather long time (over 1 h). However, during satellite signal outages additional linear velocity and altitude (depth) measurements should be applied.

### 3.2. Description of Prototype Model

To confirm the compass functionality and to estimate the expected accuracy we used a prototype model of the described integrated GNSS/IMU gyrocompass with separately installed antenna module and IMU. The prototype appearance is presented in Figure 4. The system prototype model includes

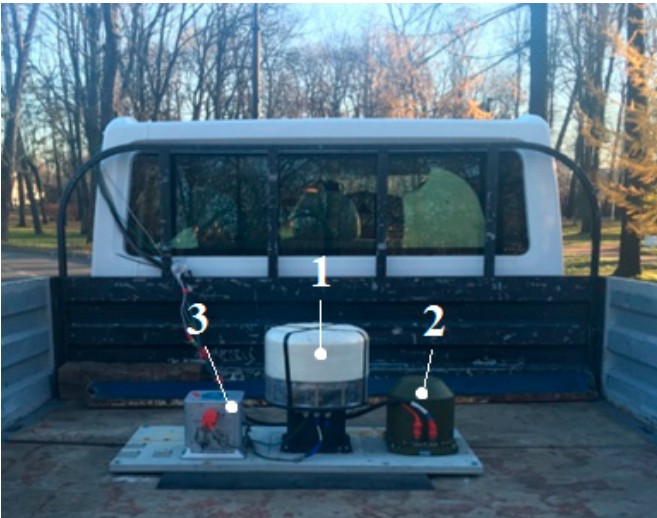

**Figure 4.** Devices installed onboard a common foundation in the car body. 1—GNSS compass, 2—IMU, 3—reference inertial navigation systems (INS).

- Compact GNSS compass with two GLONASS/GPS satellite receivers with a common clock and antenna baseline of about 0.18 m forcedly rotated by the electric drive by the harmonic law with the angle amplitude of 180° (position 1 in Figure 4);
- IMU on tactical grade FOGs (with drift level of 30°/h) and accelerometers (bias instability of 1 mg), forcedly rotated by electric drive by harmonic law with the angle amplitude of 180° (position 2 in Figure 4).

Antennas in the prototype model were spaced by 0.18 m (Figure 3b). Doubling the antenna baseline possible in the created structure can further improve the attitude accuracy accordingly. Data sampling rate for gyros, accelerometers, and angle transducers of antenna module and FOG-based IMU was 100 Hz, and for GNSS receivers, 5 Hz. During the experiment it was not needed to rotate the GNSS compass and IMU by the same algorithm. Various control laws for the drives could be used. However, GNSS compass and IMU data were synchronized at least accurate to 200 ms.

## 4. Methods

Further we describe the error models of INS and inertial sensors, models of signals measured by GNSS receivers and their errors, and the data processing algorithms.

### 4.1. INS Error Model

INS error model is described by the system of nine differential equations [5,7,9]. Errors in components of Coriolis acceleration and gravity force vector are not accounted because their contribution is negligible compared to inertial sensors' errors. These equations are given by

$$
\begin{bmatrix} \dot{\alpha} \\ \dot{\beta} \\ \dot{\gamma} \end{bmatrix} = \widetilde{\omega}_{\mathbf{n}}^{\mathbf{i},\mathbf{n}} \begin{bmatrix} \alpha \\ \beta \\ \gamma \end{bmatrix} + \begin{bmatrix} tg\varphi/R_\lambda & 0 & 0 \\ 0 & -1/R_\varphi & 0 \\ 1/R_\lambda & 0 & 0 \end{bmatrix} \delta \mathbf{V}_{\mathbf{n}}^{\mathbf{e}} + (\Omega \cos\varphi + \frac{V_E}{R_\lambda \cos^2\varphi}) \cdot \begin{bmatrix} \delta\varphi \\ 0 \\ 0 \end{bmatrix} - \mathbf{C}_{\mathbf{n},\mathbf{s}} \delta\boldsymbol{\omega}_{\mathbf{s}},
$$

$$
\delta\dot{\mathbf{V}}_{\mathbf{n}}^{\mathbf{e}} = \begin{bmatrix} f_N & 0 & -f_U \\ -f_E & f_U & 0 \\ 0 & -f_N & f_E \end{bmatrix} \begin{bmatrix} \alpha \\ \beta \\ \gamma \end{bmatrix} + \mathbf{C}_{\mathbf{n},\mathbf{s}} \delta\mathbf{f}_{\mathbf{s}}, \tag{1}
$$

$$
\delta\dot{\mathbf{p}} = \begin{bmatrix} 0 & 1/R_\varphi & 0 \\ 1/R_\lambda \cos\varphi & 0 & 0 \\ 0 & 0 & 1 \end{bmatrix} \delta\mathbf{V}_{\mathbf{n}}^{\mathbf{e}} + \begin{bmatrix} 0 \\ \frac{V_E \sin\varphi}{R_\lambda \cos^2\varphi} \\ 0 \end{bmatrix} \delta\varphi,
$$

where $\beta, \gamma, \alpha$ are pitch, roll, and heading errors; $\mathbf{p} = (\varphi, \lambda, h)^T$, $\boldsymbol{\delta}\mathbf{p} = (\delta\varphi, \delta\lambda, \delta h)^T$ is the vector of geographic coordinates including latitude, longitude and altitude, and its error vector; $\mathbf{V_n^e} = [V_E, V_N, V_U]^T$, $\boldsymbol{\delta}\mathbf{V_n^e} = [\delta V_E, \delta V_N, \delta V_U]^T$ is the vector of object linear velocity with respect to e-frame projected on n-axes and its error vector; $\overset{\smile}{\boldsymbol{\omega}}{}_\mathbf{n}^{\mathbf{i,n}}$ is a skew-symmetric matrix containing the components of n-frame angular velocity vector with respect to i-frame projected on its axes $\boldsymbol{\omega}_\mathbf{n}^{\mathbf{i,n}} = [\omega_E, \omega_N, \omega_U]^T$; $R_\lambda, R_\varphi$ are the curvature radii of the Earth normal sections; $\Omega$ is the Earth's angular velocity; $\boldsymbol{\delta}\boldsymbol{\omega}_\mathbf{s}^{\mathbf{i,s}} = [\delta\omega_{X_s}, \delta\omega_{Y_s}, \delta\omega_{Z_s}]^T$ are FOG errors in rotation angular velocity with respect to i-frame projected on s-axes; $\delta\mathbf{f}_s = [\delta f_{X_s}, \delta f_{Y_s}, \delta f_{Z_s}]^T$ are specific force errors projected on s-axes; and $\mathbf{C_{n,s}}$ is the directional cosine matrix (DCM) characterizing transition from the rotating s-frame to n-frame.

### 4.2. Error Model of Inertial Sensors

Accelerometer error model is described as a sum of biases $\boldsymbol{\delta}\mathbf{f_{0s}}$ described by Wiener processes (random walk) with generating noises $\mathbf{w_{f_0}}$ and measurement noises $\mathbf{w_f}$, i.e.,

$$\begin{aligned} \delta\mathbf{f}_s &= \boldsymbol{\delta}\mathbf{f_{0s}} + \mathbf{w_f}, \\ \dot{\boldsymbol{\delta}\mathbf{f}}_{\mathbf{0s}} &= \mathbf{w_{f_0}}. \end{aligned} \tag{2}$$

Gyro error model in s-frame axes bound with the inertial sensor axes is set as follows, with account for IMU rotation about vertical axis $Z_s$:

$$\begin{aligned} \delta\omega_{X_s} &= \delta\omega_{0X_s} + \delta\omega_{RX_s} + w_{gx}, \\ \delta\omega_{Y_s} &= \delta\omega_{0Y_s} + \delta\omega_{RY_s} + w_{gy}, \\ \delta\omega_{Z_s} &= \delta\omega_{0Z_s} + \delta S_{gY_s}\omega_{Z_s} + w_{gz}, \\ \boldsymbol{\delta}\dot{\boldsymbol{\omega}}_0 &= \mathbf{w}_{\boldsymbol{\omega}_0}, \\ \delta\dot{S}_{gY_s} &= w_{S_g}, \end{aligned} \tag{3}$$

where $\boldsymbol{\delta}\boldsymbol{\omega}_0 = [\delta\omega_{0X_s}, \delta\omega_{0Y_s}, \delta\omega_{0Z_s}]^T$ are the gyro biases characterizing instability from run to run and in run and presented by Wiener processes with generating noises $\mathbf{w}_{\boldsymbol{\omega}_0}$; $\delta S_{gZ_s}$ is the scale factor error of the gyro with measurement axis parallel to $Z_s$ rotation axis with generating noise $w_{S_g}$; $\delta\omega_{Ri}(i = X_s, Y_s)$ are the rhumb drifts of the gyros with measurement axes orthogonal to the rotation axis; and $\omega_{Z_s}$ is the angular velocity projected on IMU and antennas' rotation axis $Z_s$.

Rhumb drifts are the components of gyro bias whose projections on geographical axes *ENU* remain constant. These error components may arise due to the effect of the Earth's magnetic field to which FOGs are sensitive [41]. FOG rhumb drifts cannot be compensated by IMU rotation through angle $u$ and increase the heading error. Rhumb drifts are presented by the first harmonic of Fourier expansion of heading angle and rotation angle u

$$\begin{aligned} \Delta\omega_{RX_s} &= A_{X_s}\cos(K - u) + B_{X_s}\sin(K - u), \\ \Delta\omega_{RY_s} &= A_{Y_s}\cos(K - u) + B_{Y_s}\sin(K - u), \\ \left[A_{X_S}, B_{X_S}, A_{Y_S}, B_{Y_S}\right]^T &= \mathbf{w_R}, \end{aligned} \tag{4}$$

where $A_i, B_i(i = X_S, Y_S)$ are the amplitudes of rhumb drift components for the gyros with measurement axes orthogonal to the rotation axis presented by the relevant Wiener processes; $\mathbf{w_R}$ is a four-dimensional vector of generating noises.

### 4.3. Models of Satellite Signals and Their Errors

As raw data for GNSS/INS data joint processing we use pseudorange $\rho^i$, radial velocity $\dot{\rho}^i$, and single difference of carrier phase $\Delta\phi^i$ received by the spaced antennas for each observed i-th

satellite. The measured pseudorange $\rho^{iGNSS}$ and radial velocity $\dot{\rho}^{iGNSS}$ of reference antenna for each observed *i*-th satellite of *j*-th GNSS, $j = GPS, GLONASS$, are described by

$$
\begin{aligned}
\rho^{iGNSS} &= \rho^i + c\delta T^j + \varepsilon_\rho, \\
\dot{\rho}^{iGNSS} &= \dot{\rho}^i + \frac{d(c\delta T^j)}{dt} + \varepsilon_{\dot{\rho}},
\end{aligned}
\tag{5}
$$

where $\rho^i$ is the true distance between the satellite and the receiver, $\dot{\rho}^i$ is the true radial velocity, *c* is the light velocity, $\delta T^j$, $j = GPS, GLONASS$ is the receiver clock error depending on the GNSS, $\varepsilon_\rho, \varepsilon_{\dot{\rho}}$ are the noises of GNSS receivers. Values $\delta\rho^j = c\delta T^j$ and $\delta v^j = \frac{d(c\delta T^j)}{dt}$ are the clock frequency shifts and velocity of their changeability with respect to j-th GNSS satellite data described by the following model:

$$
\begin{aligned}
\dot{\delta\rho}^j &= \delta v^j + w^j_{\delta\rho}, \\
\dot{\delta v}^j &= K^j + w^j_{\delta\dot{\rho}}, \\
\dot{K}^j &= w^j_K,
\end{aligned}
\tag{6}
$$

$w^j_{\delta\rho}, w^j_{\delta\dot{\rho}}, w^j_K$ are zero-mean discrete white noises with the known variances. Factor $K^j$ characterizes the clock frequency shift and is described by Wiener process.

The measured single difference of i-th satellite carrier phase received by the spaced antennas $L^i\Delta\phi^{iGNSS}$ was preliminary processed. Values—calculated as the difference between the phase difference determined at the first step for the current satellite at the current epoch and the antenna baseline with further rounding to the integer—were excluded from $L^i\Delta\phi^{iGNSS}$. After that the measurements $L^i\Delta\phi^{iGNSS}$ can be written as

$$
L^i\Delta\phi^{iGNSS} = \Delta\rho^i + \delta S f^i + \varepsilon_{L\Delta\phi},
\tag{7}
$$

where $\Delta$ is the operator denoting the single difference, $L^i$ is the carrier wavelength, $\delta C f^i$ are the errors conditioned by the residual ambiguity of phase single differences for the satellite, $\varepsilon_{L\Delta\phi}$ is GNSS receiver errors in the phase carrier measurement (including the multipath errors). Application of two satellite navigation receivers with a common clock excludes the receivers' clock error from the measurements [6].

### 4.4. Filtering Algorithms Used for GNSS/IMU Gyrocompass Measurements

Below we briefly describe the filtering algorithms used in GNSS/IMU gyrocompass referring to the block diagram presented in Figure 5. INS comprised in GNSS/IMU gyrocompass (block INS in Figure 5) uses rotation angular velocity $\boldsymbol{\omega_s^{i,s}}$ and specific force $\mathbf{f}_s$ to generate the geographic coordinates $\mathbf{p} = (\varphi, \lambda, h)^T$, projections of linear velocity with respect to e-frame on n-frame $\mathbf{V_n^e} = [V_E, V_N, V_U]^T$, and attitude of b-frame with respect to n-frame. Euler-Krylov angles: heading *K*, roll $\theta$ and pitch $\psi$ are used as attitude parameters. Equations of INS operation are detailed in references [5,7,9]. To start the INS algorithms at the moment of GNSS/IMU gyrocompass turn-on, navigation solution from GNSS receivers can be used if the sufficient number of navigation satellites is observed (minimum 4 satellites of one GNSS or 5 satellites of two different GNSS).

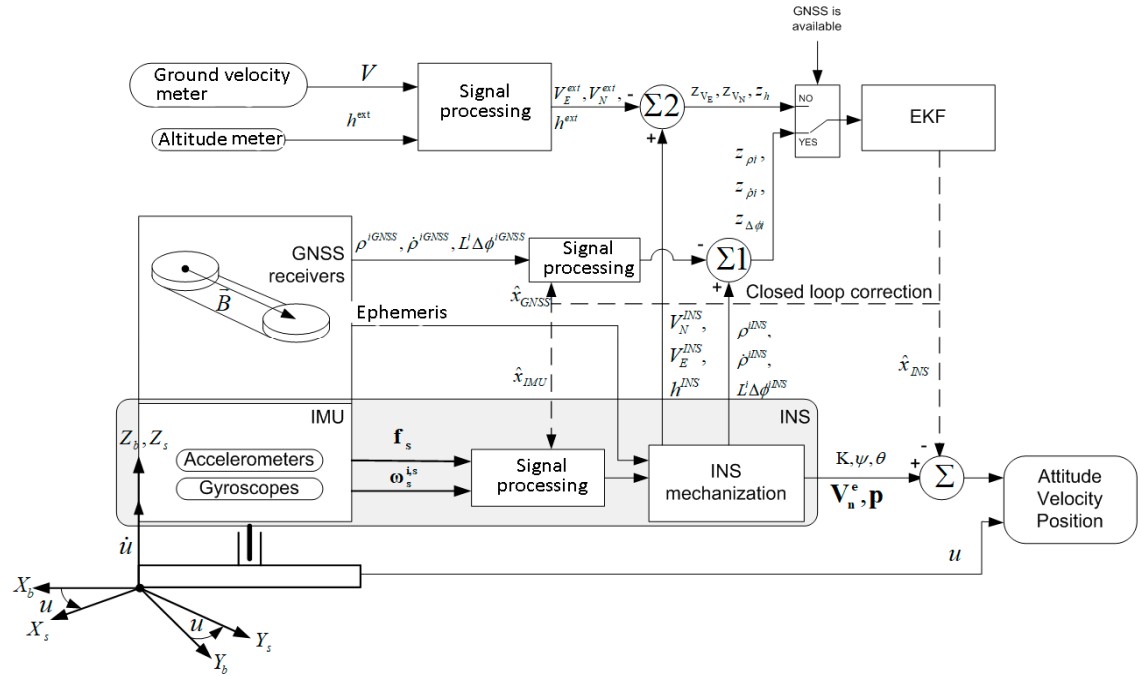

**Figure 5.** Block diagram of GNSS/IMU gyrocompass algorithms.

If satellite data are available, the state vector including INS errors ($x_{INS}$), components of inertial sensors' error models ($x_{IMU}$), and GNSS receiver errors ($x_{GNSS}$) is estimated. In this case the system state vector is given by

$$x = \begin{bmatrix} x_{INS} \\ x_{IMU} \\ x_{GNSS} \end{bmatrix},$$

$$x_{INS} = \begin{bmatrix} \alpha & \beta & \gamma & \delta\mathbf{V_n^e}^T & \delta\mathbf{p}^T \end{bmatrix}^T,$$

$$x_{IMU} = \begin{bmatrix} \delta\mathbf{f_0}^T & \delta\boldsymbol{\omega_0}^T & \delta S_{gZ_s} & A_{X_s} & B_{X_s} & A_{Y_s} & B_{Y_s} \end{bmatrix}^T,$$

$$x_{GNSS} = \begin{bmatrix} \delta\rho^j, \delta v^j, K^j, \delta Cf^i \end{bmatrix}^T.$$

(8)

Parameters generated by INS and ephemeris data are used to generate the calculated pseudorange $\rho^{iINS}$, radial velocity $\dot{\rho}^{iINS}$ and the first phase difference in length units for each observed i-th satellite in rotating frame $X_s Y_s Z_s$—$L^i \Delta\phi^{iINS}$ determined as

$$L^i \Delta\varphi^{iINS} = \left|\vec{B}\right| (\mathbf{s_n^i})^T (\mathbf{C_{n,b}} \mathbf{C_{b,s}} \mathbf{b_s}),$$

(9)

where $L^i$ is the carrier wavelength; $\mathbf{s_n^i} = \begin{pmatrix} s_E^i & s_N^i & s_U^i \end{pmatrix}^T$ is the unit vector setting the direction to the i-th satellite in n-frame and its components; $\mathbf{b_s} = \begin{pmatrix} b_{X_S} & b_{Y_S} & b_{Z_S} \end{pmatrix}^T$ is the unit vector of antenna baseline in s-frame; $\mathbf{C_{n,b}}$ is the transformation matrix from b-frame to n-frame and containing heading, pitch, and roll data; $\mathbf{C_{b,s}}$ is the transformation matrix from s-frame to b-frame and calculated by the current rotation angle $u$.

The filtering measurements are generated as differences $z_{\rho i}, z_{\dot{\rho} i}, z_{\Delta\phi i}$ between the calculated $\rho^{iINS}$, $\dot{\rho}^{iINS}, L^i \Delta\phi^{iINS}$ and measured $\rho^{iGNSS}, \dot{\rho}^{iGNSS}, L^i \Delta\phi^{iGNSS}$

$$z_{\rho i} = \rho^{iINS} - \rho^{iGNSS},$$

(10)

$$z_{\dot{\rho}i} = \dot{\rho}^{iINS} - \dot{\rho}^{iGNSS}, \tag{11}$$

$$z_{\Delta\varphi i} = \left( L^i \Delta\varphi^{iINS} - L^i \Delta\varphi^{iGNSS} \right)/\left|\vec{B}\right|. \tag{12}$$

Assuming that inertial sensors determine the initial rotation angles at least accurate to 5°, measurements (10)–(12) can be presented in linearized form:

$$\widetilde{z}_{\rho i} = \left( M_1 \cdot \left[ \frac{\partial \rho_i}{\partial x_e}, \frac{\partial \rho_i}{\partial y_e}, \frac{\partial \rho_i}{\partial z_e} \right]^T \right)^T \cdot \delta\mathbf{p} - \delta\rho + \varepsilon_\rho, \tag{13}$$

$$\widetilde{z}_{\dot{\rho}i} = \left( M_1 \cdot \left[ \frac{\partial \dot{\rho}_i}{\partial x_e}, \frac{\partial \dot{\rho}_i}{\partial y_e}, \frac{\partial \dot{\rho}_i}{\partial z_e} \right]^T \right)^T \cdot \delta\mathbf{p} + \left( M_2 \cdot \left[ \frac{\partial \dot{\rho}_i}{\partial \dot{x}_e}, \frac{\partial \dot{\rho}_i}{\partial \dot{y}_e}, \frac{\partial \dot{\rho}_i}{\partial \dot{z}_e} \right]^T \right)^T \left[ \begin{array}{c} \delta\mathbf{V_n^e} \\ \delta\mathbf{p} \end{array} \right] - \delta v + \varepsilon_{\dot{\rho}}, \tag{14}$$

$$\widetilde{z}_{\Delta\phi i} \cong \left( b_N \cdot s_E^i - b_E \cdot s_N^i \right)\alpha + \delta C f^i + v_{zi}, \tag{15}$$

where $[x_e, y_e, z_e]$, $[\dot{x}_e, \dot{y}_e, \dot{z}_e]$ are the Cartesian coordinates and the components of relative linear velocity vector projected on e-frame, $v_{zi}$ are the measurement errors including mainly GNSS receiver errors in carrier phase (including multipath errors) and fluctuation errors in attitude of antenna baseline unit vector $\vec{b}$ with respect to IMU axes (for example due to the vibrations). Matrices $M_1$ and $M_2$ contained in (13) and (14) are given by

$$M_1 = \left[ \begin{array}{ccc} -R_\lambda s\varphi c\lambda & -R_\lambda s\varphi s\lambda & R_\lambda c\varphi \\ -R_\lambda c\varphi s\lambda & R_\lambda c\varphi c\lambda & 0 \\ c\varphi c\lambda & c\varphi s\lambda & s\varphi \end{array} \right],$$

$$M_2 = \left[ \begin{array}{ccc} -s\lambda & c\lambda & 0 \\ -s\varphi c\lambda & -s\varphi s\lambda & c\varphi \\ c\varphi c\lambda & c\varphi s\lambda & s\varphi \\ -(V_N c\varphi c\lambda + V_H s\varphi c\lambda) & -(V_N c\varphi s\lambda + V_H s\varphi s\lambda) & (-V_N s\varphi + V_H c\varphi) \\ (-V_E c\lambda + V_N s\varphi s\lambda - V_H c\varphi s\lambda) & (-V_E s\lambda - V_N s\varphi c\lambda + V_H c\varphi c\lambda) & 0 \\ 0 & 0 & 0 \end{array} \right]. \tag{16}$$

Here $s\varphi, s\lambda, c\varphi, c\lambda$ denote $\sin\varphi, \sin\lambda, \cos\varphi, \cos\lambda$, respectively.

To estimate the state vector (8) using differential measurements (10)–(12), extended Kalman filter is applied (block EKF in Figure 4) with feedbacks over the complete state vector $x$. Its estimated components $\hat{x}_{INS}, \hat{x}_{IMU}, \hat{x}_{GNSS}$ generated by EKF are used to update INS output and accelerometer, gyro, and satellite measurements under no failures of GNSS receivers and satellite visibility ("Closed loop correction" in Figure 5). This actually defines the mode with feedbacks over the complete state vector.

As seen from measurements (15), heading can be updated using phase measurements from only one satellite, which would be confirmed by experimental results. The residual phase ambiguity $\delta C f^i$ is excluded due to rotation because antenna baseline orientation with respect to satellite direction changes, and gyro data are applied. Note that even with significant INS heading errors, filtering using differential phase measurements (15) by data from a single satellite is sufficient. Then using a common clock is very important, which excludes the clock error of satellite navigation receivers from measurements (15).

In filtering algorithms, rejection of unreliable measurements is very important. Further we describe the algorithms applied to reject phase measurements with enhanced errors and to compensate the effect of low-frequency noise in the measured single phase differences for each satellite occurring randomly for instance due to multipath. When the satellite number was changed in the receiver channel, in EKF covariance equations the error variance of phase ambiguity $\delta C f^i$ relevant for this channel was restored to the initial value, and the previous estimate was zeroed. During the processing

of phase measurements from each visible satellite, measurements with enhanced errors were revealed according to the criterion generated using the EKF covariance:

$$\sigma_{z\Delta\phi i(k+1)} = \sqrt{diag\left(H_{k+1}P_{k/k+1}H_{k+1}^T + R_{k+1}\right)},$$
$$\left|\widetilde{z}_{\Delta\phi i}\right| \le k_d \sigma_{z\Delta\phi i(k+1)}, \tag{17}$$

where $\sigma_{z\Delta\phi i(k+1)}$ are the calculated RMS errors of measurements $\widetilde{z}_{\Delta\phi i}$; $P_{k/k+1}$ is the predicted EKF covariance matrix; $H_{k+1}$ is the measurement matrix at step k+1; $R_{k+1}$ is the measurement noise covariance matrix; $k_d = 4...6$. These relations are given for the composite vector (8) with account for $x_{GNSS} = [\delta\rho^j, \delta v^j, K^j, \delta C\dot{f}^j]^T$ and for measurements (10)–(12).

If $\left|\widetilde{z}_{\Delta\phi i}\right| > k_d\sigma_{z\Delta\phi i(k+1)}$, GNSS receiver measurements for the i-th satellite were not used.

Additionally, to reduce the effect of perturbed phase measurements on the system errors, for each satellite the EKF was restarted using the relevant estimate $\delta\hat{C}f^i$. The following criterion was used:

$$\left|L_{kfi}\right| < L_{kfi}^{dop}, \tag{18}$$

where $L_{kfi}$ is the value of residuals $L_{ki} = \widetilde{z}_{\Delta\phi i}\left|\vec{b}\right|$ of differential phase measurements smoothed by the low-pass filter as a first-order with a time constant equal to the period of antenna module modulation rotation; $L_{kfi}^{dop}$ is the acceptable value of smoothed residuals. With $\left|L_{kfi}\right| \ge L_{kfi}^{dop}$ the current measurement $\widetilde{z}_{\Delta\phi i(k+1)}$ was not used, and the Kalman filter was restarted.

In case of lost or ineffective satellite signals the relevant signal is generated in the switch "GNSS is available" (Figure 5). Then INS is corrected by additional external measurements of linear velocity and altitude (depth) $h^{ext}$, and filtering is performed for the state vector including only INS errors ($x_{INS}$) and components of inertial sensors' error models ($x_{IMU}$). With no GNSS signals, the system state vector is given by

$$x = \begin{bmatrix} x_{INS} \\ x_{IMU} \end{bmatrix},$$
$$x_{INS} = \begin{bmatrix} \alpha & \beta & \gamma & \delta\mathbf{V_n^e}^T & \delta\mathbf{p}^T \end{bmatrix}^T, \tag{19}$$
$$x_{IMU} = \begin{bmatrix} \delta\mathbf{f}_0^T & \delta\boldsymbol{\omega}_0^T & \delta S_{gZ_s} & A_{X_s} & B_{X_s} & A_{Y_s} & B_{Y_s} \end{bmatrix}^T.$$

External data can be generated using absolute or relative velocity sensors, and barometric altitude (depth) sensor. Note that rotation of inertial sensors and applying tactical grade FOGs allows heading determination with the acceptable accuracy for a long time after satellite signal loss. However it should be noted that the previous period of fault-free satellite signal reception needs to be rather long to have enough time to estimate all the components of inertial sensor error models. As seen further from the experimental results, this period can reach 1500 s. In this mode, filtering is realized using differential measurements $z_{V_E}, z_{V_N}, z_h$ generated in unit $\Sigma2$ (Figure 4) as a difference between the estimated $V_N^{INS}, V_E^{INS}, h^{INS}$ and measured $V_N^{ext}, V_E^{ext}, h^{ext}$ values:

$$z_{V_E} = V_E^{INS} - V_E^{ext},$$
$$z_{V_N} = V_N^{INS} - V_N^{ext}, \tag{20}$$
$$z_h = h^{INS} - h^{ext}.$$

If we use the sensors of object velocity relative to the environment (water or air) $V_L$ projected onto the longitudinal axis, velocity measurements can be rewritten as follows:

$$V_E^{ext} = V_L \sin K, \quad V_N^{ext} = V_L \cos K,$$
$$\widetilde{z}_{V_E} = \Delta V_E - V_N\alpha + V_{TE} - \nu_{V_E}, \tag{21}$$
$$\widetilde{z}_{V_N^L} = \Delta V_N + V_E\alpha + V_{TN} - \nu_{V_N},$$

where $V_{TE}, V_{TN}$ are the Eastern and Northern components of drift or current velocity, respectively; and $v_{Vi}(i = E, N)$ are the measurement errors including instrumental errors approximated by discrete white noises.

If we use two-component sensors of ground velocity $V_{X_b}, V_{Y_b}$ (such as Doppler speed logs or acoustic logs), velocity measurements take the form

$$
\begin{aligned}
(\mathbf{V_n^e})^{ext} &= C_{n,b} \cdot \left[ V_{X_b}, V_{Y_b}, 0 \right]^T, \\
\widetilde{z}_{V_E} &= \Delta V_E - V_N \alpha - v_{V_E}, \\
\widetilde{z}_{V_N^L} &= \Delta V_N + V_E \alpha - v_{V_N}.
\end{aligned}
\tag{22}
$$

The measurements in altitude (depth) are generated as

$$
z_h = h^{INS} - h^{ext} \cong \Delta h + v_h,
\tag{23}
$$

where $v_h$ is the error of external altitude (depth) sensor. Estimates of state vector components $\hat{x}_{INS}$, $\hat{x}_{IMU}$ are applied to update INS output, and to update gyro and accelerometer data with no satellite signals ("Closed loop correction" in Figure 4).

## 5. Results

To confirm the compass operability and to estimate the expected accuracy we used a prototype model of the described GNSS/IMU gyrocompass (see Section 3.2) installed in a car body (Figure 4). The tests were conducted in urban conditions (St. Petersburg, Russia), with significant distortions in GNSS signal propagation.

The following data arrays were generated in the prototype tests: inertial sensors' data $\mathbf{f}_s, \boldsymbol{\omega}_s^{i,s}$; IMU rotation angle $u$; pseudorange $\rho^i$; radial velocity $\dot{\rho}^i$; and single phase difference of carrier $\Delta \phi^i$ received by the spaced antennas for each observed i-th satellite, ephemeris data, and external measurements of linear velocity $V_N^{ext}, V_E^{ext}$. The obtained arrays were postprocessd using algorithms described in Section 4.3 implemented in MATLAB (Simulink). It should be mentioned that for the measurements' processing initial heading error in each system start was taken to be 100°. During data integration we simultaneously used maximum five satellite signals (GPS or GLONASS) to generate phase measurements (12) so that to reduce the computation burden. To generate pseudorange (10) and radial velocity (11) measurements we applied the signals of the whole observable satellite constellation (both GPS and GLONASS).

The car test path is shown in Figure 6. The figure presents relative displacement coordinates ($\Delta$Fi, $\Delta$La) in meters calculated from the route initial point.

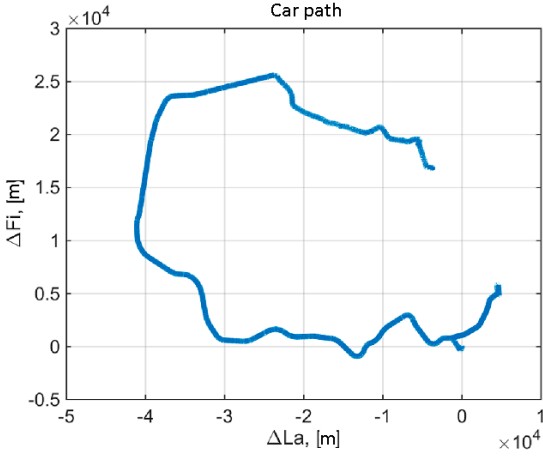

**Figure 6.** Car path during the tests.

Reference attitude parameters were obtained using the data from reference INS (position 3 in Figure 4) with heading generation accuracy of $6'\sec\varphi$. Heading and heading rate during the route are presented in Figure 7 (ovals denote the maneuvers).

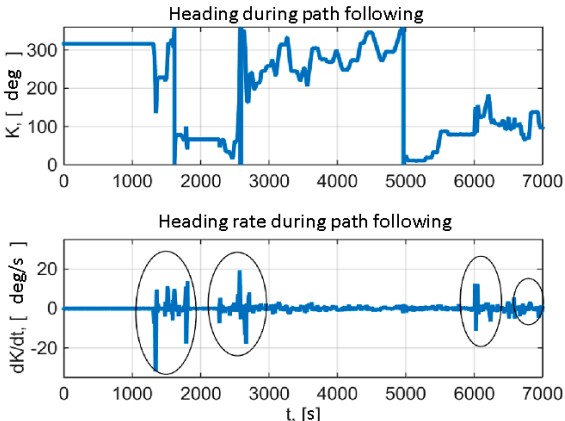

**Figure 7.** Heading and heading rate during the route.

Below the following results are presented.

Section 5.1 demonstrates the results with GNSS signals from multiple satellites. The state vector (8) was estimated using differential measurements (10)–(12) including pseudorange, radial velocity, and phase single difference.

Section 5.2 demonstrates the results with a signal of a single GLONASS satellite suitable for phase measurements generation (12) and with further signal outage. It is shown that with phase measurements (12) of a single satellite in filtering problem, all the components of FOG error model are estimated (3). Then the heading accuracy is slightly degraded as compared with the use of phase data from five satellites. Note that pseudorange (10) and radial velocity (11) measurements were generated by the signals from minimum five satellites.

Section 5.3 presents the results under total GNSS signal outage. The filtering problem then uses differential measurements (20) generated by data from additional external linear velocity sensors. It is shown that acceptable heading accuracy (max error of 3°) can be provided for a long time.

*5.1. Results with Phase Measurment Signals from Multiple Satellites*

Consider the results under good visibility of GNSS signals from multiple satellites obtained using differential measurements of pseudorange, radial velocity, and phase single difference (10)–(12) to estimate the state vector (8). Here and below phase measurements (12) from minimum five satellites with elevations from 30° to 75° are used. By the satellite elevation we mean the angle counted from the horizon plane to satellite direction. Priority was given to the lowest satellites from this range because their phase measurements are more effective for heading updates: factor of angle $\alpha$ in (15) grows drastically. The lower bound of elevation range was selected depending on the prevailing height of city buildings to reduce the multipath effects.

Numbers of GPS satellites used for attitude determination in five channels of satellite receivers are presented in Figure 8. It can be seen that they often change in the receiver channels. Figure 9 presents the elevations and azimuths of satellites with these numbers. Asterisk shows the moment when the i-th satellite appears in visibility zone during path following. The data of i-th satellite started to be used as soon as it entered the allowed elevation zone (red solid lines in Figure 9).

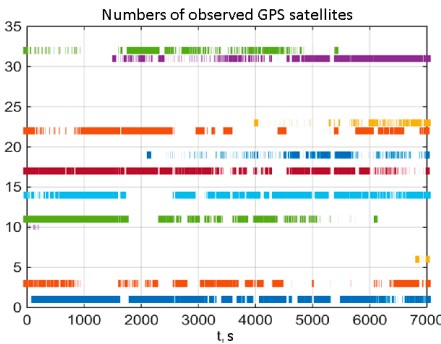

**Figure 8.** Numbers of GPS satellites in five channels of satellite receivers.

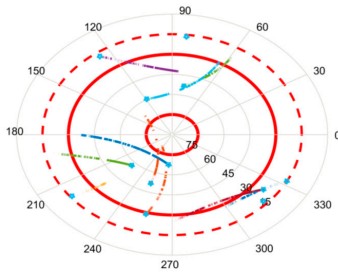

**Figure 9.** Azimuth and elevation of GPS satellites. Red solid lines show the acceptable elevation zones.

Figure 10a,b present heading errors during the stop and during motion using phase measurements from five GPS satellites. Here and below, heading error was determined compared to reference INS. As seen from the figures, the heading error after the completion of transients in the system including the heating reaches 1.2° and 1° for the stop (Figure 10a) and in motion (Figure 10b), respectively. Figure 10c present the heading error histogram during motion in a steady-state time interval between two vehicle manoeuvres.

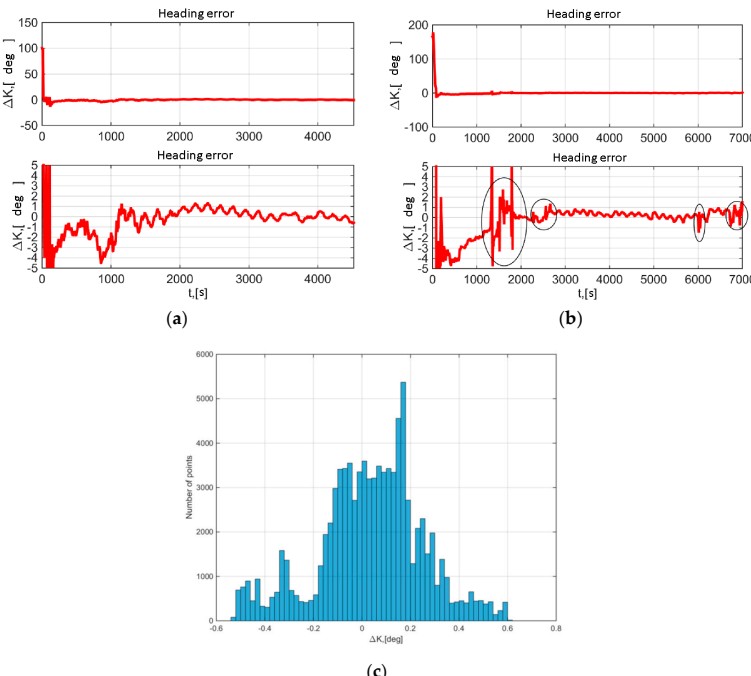

**Figure 10.** Heading errors using data from five GPS satellites: (**a**) During the stop; (**b**) During motion (In lower plots, the ordinate axis is zoomed in); (**c**) The heading error histogram.

The estimated components of FOG error model (3): rhumb drifts and biases using differential measurements (10)–(12) are shown in Figures 11 and 12, respectively.

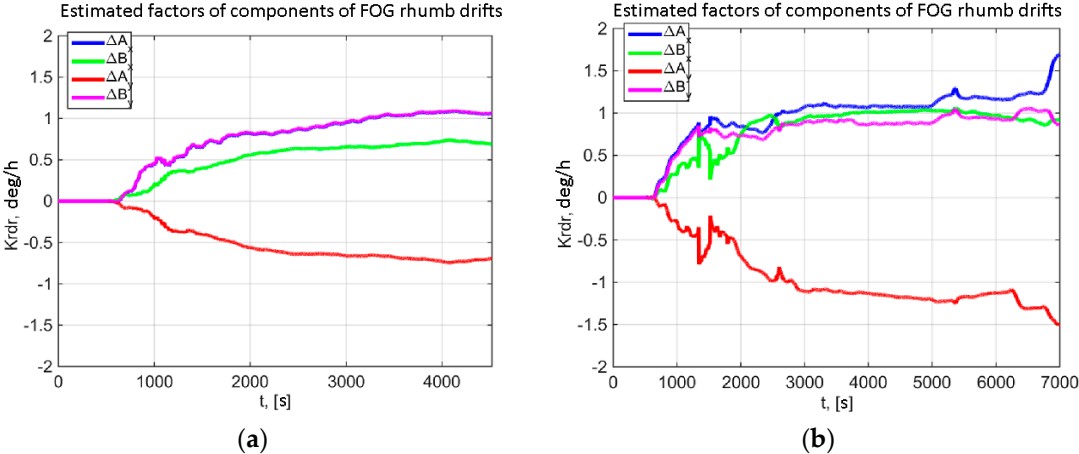

**Figure 11.** Estimated FOG rhumb drifts: (**a**) During the stop; (**b**) During the car motion.

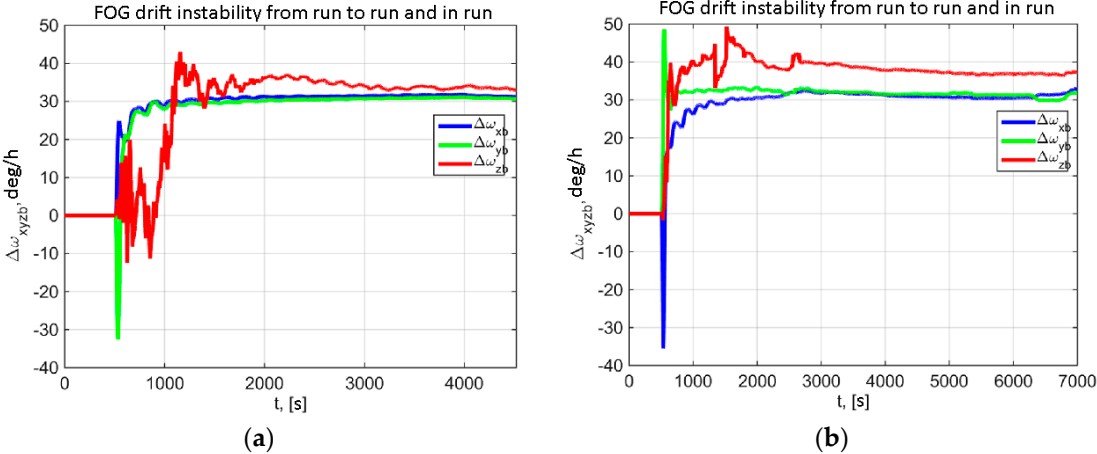

**Figure 12.** Estimated FOG bias: (**a**) During the stop; (**b**) During the car motion.

*5.2. Results with Phase Measurements from a Single Satellite*

Now consider the results obtained if phase measurements (12) from a single GLONASS satellite are used in the filtering problem. The pseudorange (10) and radial velocity (11) measurements are used from minimum five satellites.

Figure 13 presents the heading errors, and Figure 14, phase residuals for Kalman filtering using phase measurements (12) of a single GLONASS satellite from the elevation range 30°–75°. As seen from the figures, the heading error in steady mode does not exceed 1.5° (without account for errors in synchronization with the reference INS).

Figure 14 presents the phase residuals used in the algorithm for rejection of unreliable phase measurements. The values of residual $L_{kfi}$ are given for one GLONASS satellite under vibrations during the car motion. Figure 14 clearly shows distinctive periodic perturbations (with periods of units of minutes, indicated by marker "1"), whose character suggests their connection with GNSS signal rereflections from the surrounding objects.

Figure 15 below shows FOG estimated biases and rhumb drifts in these conditions.

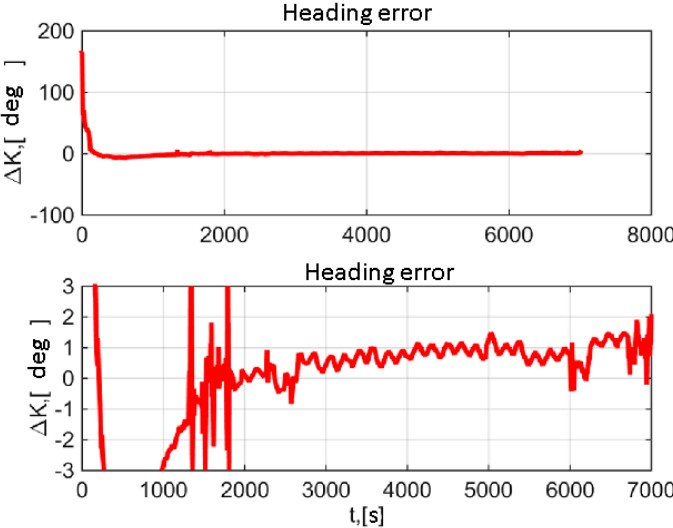

**Figure 13.** Heading errors during car motion using phase measurements from a single GLONASS satellite.

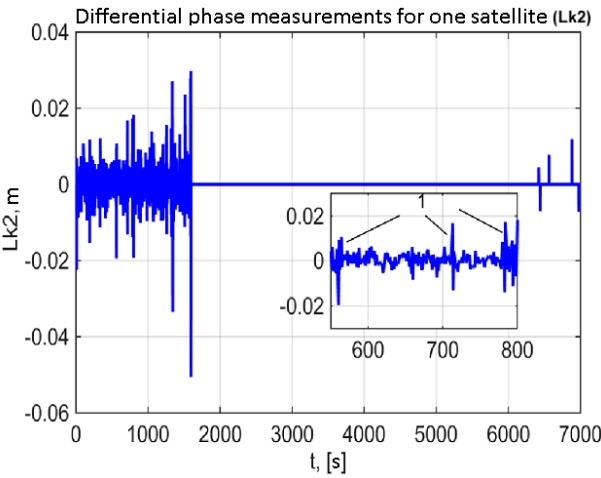

**Figure 14.** Phase residuals during car motion using phase measurements from a single GLONASS satellite.

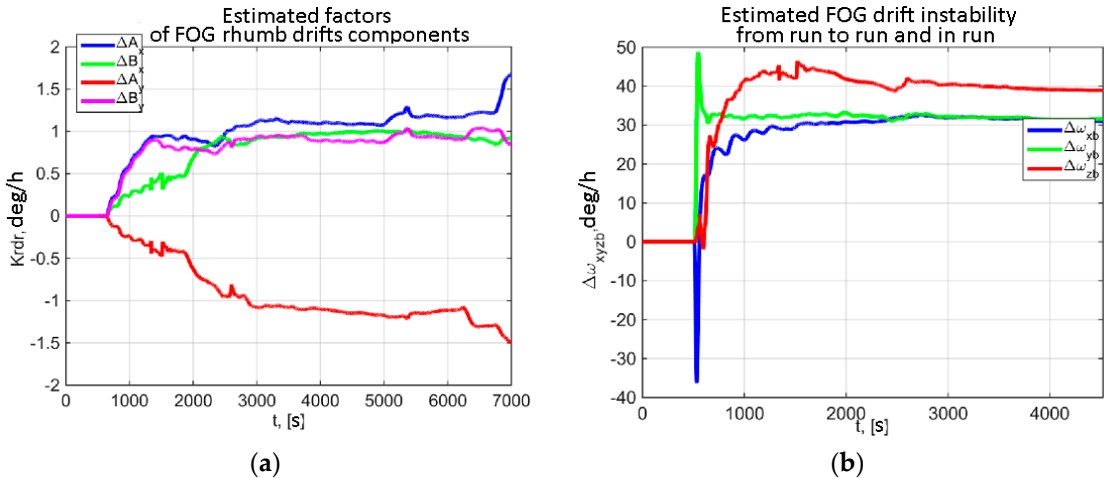

**Figure 15.** Estimated parameters of FOG error model using phase measurements from a single GLONASS satellite: (**a**) Rhumb drift; (**b**) Bias.

Comparison of Figures 11, 12 and 15 reveals that even using effective phase measurements from a single GLONASS satellite FOG bias and rhumb drift can be effectively estimated.

## 5.3. Results with Complete GNSS Signal Outage

It would be also interesting to estimate how accurately the prototype GNSS/IMU gyrocompass after complete loss of satellite signals.

There were no complete satellite signal outages during the car test, so it was decided to use data obtained at the previous point with forced removal of all satellite signals from processing starting from 2001 s. The following operation mode was used. GNSS/IMU gyrocompass was started using phase measurements (12) of a single effective GLONASS satellite and measurements (10), (11) of the whole observable satellite constellation. The whole state vector (8) including FOG biases and rhumb drifts was estimated for 2000 s. Further all satellite signals were forcedly removed from processing, and INS was corrected by additional external linear velocity measurements. The altitude during the motion was considered identically zero, which is acceptable for automobile conditions. In this case the state vector (19) was estimated by measurements (20). But the accuracy of estimating FOG bias and rhumb drift in this case is much lower than when using measurements (10), (11).

Figure 16 presents heading, roll and pitch errors. The experimental results demonstrate that with no satellite signals the maximum heading error does not exceed 3° for 1.5 h after signal outage. Maximum roll and pitch errors did not exceed 0.2°, which is typical for all conditions of car test. These errors are given without account for synchronization errors between GNSS/IMU gyrocompass prototype and reference INS, which critically degrade the attitude error during the maneuvers (shown by ovals in Figure 16b). Figure 16c present the heading error after signal outage between two vehicle manoeuvres.

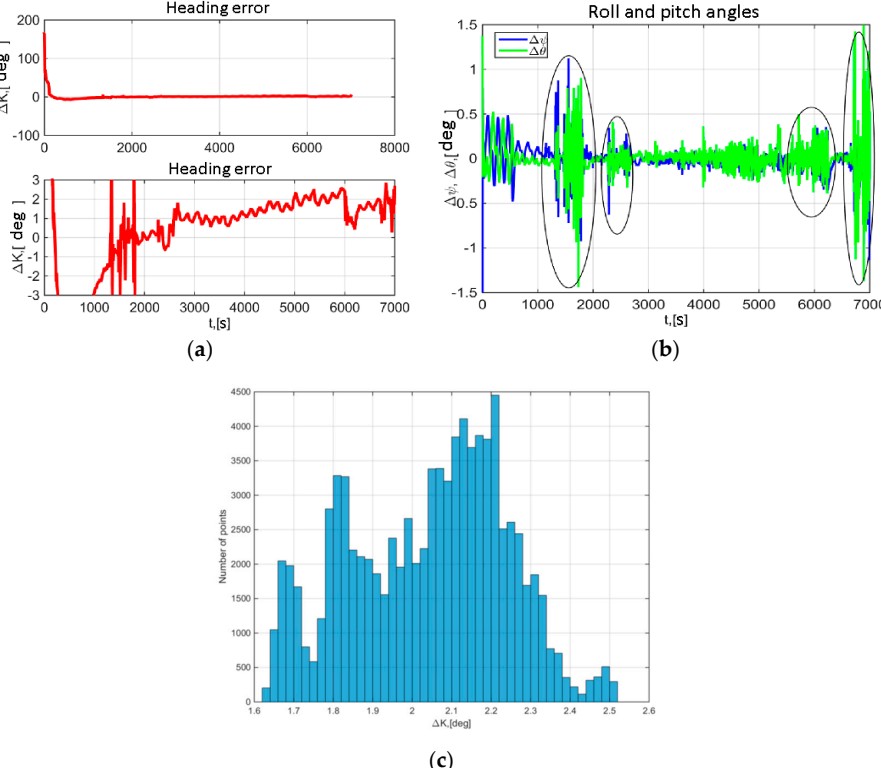

**Figure 16.** Attitude error of GNSS/IMU gyrocompass with satellite signals completely removed from processing starting from 2000 s: (**a**) Heading error; (**b**) Roll and pitch errors; (**c**) The heading error histogram.

## 6. Discussion

First of all let us analyze the heading accuracy under good visibility of GNSS signals from multiple satellites. As seen from the Figure 10, the heading error after the completion of transients in the system including the heating (at 1000–1500 s) reaches 1.2° and 1° for the stop (Figure 10a) and in motion (Figure 10b), respectively. This error level is observed without account for synchronization errors between INS and GNSS/IMU gyrocompass prototype. Synchronization error was 100–200 ms and critically degraded the heading error during the maneuver (Figure 10b, shown by an oval). The difference between heading errors during the stop and during motion is assumed to be conditioned mainly by multipath effects. As is known [42], the rate of change of multipath error is directly proportional to the distance between the receiver and the obstruction from which the signal is reflected. During the stop, multipath error becomes a low-frequency one, which complicates its filtering using IMU data. Plots in Figure 10 show that the transient mode (heading determination with error of 1–1.5°) lasts 1200–1500 s after GNSS/IMU gyrocompass start. This duration of alignment is conditioned by several factors. First, large FOG biases and rhumb drifts should be estimated. Second, since horizon alignment of the system should be performed, to shorten the transient process during FOG drift estimation a delay up to 500 s from the system start was introduced, during which the drifts were not estimated (see Figure 11). The third reason is the system self-heating period. The plots in Figure 12 demonstrate that FOG rhumb drift reaches 1.5 deg/h, and FOG bias reaches 30–40 deg/h.

As for heading errors for phase measurements (12) from only a single GLONASS satellite it is easy to see by comparison of Figure 13 (single satellite) and Figure 10b (five satellites) that even using phase measurements from a single GLONASS satellite the heading error does not greatly differ from the error with five satellites available: 1.5° vs 1°. The duration of transient process is also comparable. From the presented results (Figure 13) it follows that the considered prototype GNSS/IMU gyrocompass preserves the acceptable heading accuracy with phase measurements from a single satellite.

Finally it is important to analyze the heading error after complete loss of satellite signals. The presented results (Figure 16) show that GNSS/IMU gyrocompass can operate with the acceptable heading accuracy even with a long satellite signal outage (up to 1.5 h): the error is max 3°. In this case the duration of GNSS/IMU gyrocompass performance with the required accuracy will be determined mainly by FOG error instability. In its turn FOG bias and rhumb drift can be effectively estimated during the system alignment using effective phase measurements from a single GLONASS satellite. The experimental results demonstrate maximum roll and pitch errors did not exceed 0.2° for all conditions of car test.

## 7. Conclusions

The results of automobile tests have confirmed the operability of the GNSS/IMU gyrocompass prototype. The system provided max heading error within ±1.5°, and roll and pitch errors, within ±0.2° in steady mode during the car motion in urban conditions.

It is shown that with the use of a single-axis rotating platform with installed IMU and two satellite antennas with two satellite navigation receivers with a common clock the proposed GNSS/IMU gyrocompass provides the heading solution based on phase measurements from a single satellite.

It is noted that IMU rotation and application of tactical grade FOGs helps to keep the heading, roll and pitch accuracy during long GNSS signal outages if INS is updated by additional external linear velocity and altitude measurements. The test results demonstrate that GNSS/IMU gyrocompass provides heading solution with error of max 3° for 1.5 h after satellite signal loss. The duration of GNSS/IMU gyrocompass performance with the required accuracy is determined mainly by errors in estimation of FOG bias and rhumb drifts realized in the presence of satellite signals.

**Author Contributions:** Conceptualization, G.E.; methodology, G.E., M.E., and A.S.; software, A.S. and B.B.; investigation, B.B.; resources, M.E. and D.E.; writing—original draft preparation, E.D. and A.S.; writing—review and editing, O.S., and D.E.; visualization, E.D.; supervision, O.S.; project administration, D.V. All authors have read and agreed to the published version of the manuscript.

**Funding:** This work was financially supported by Government of Russian Federation (Grant 08-08).

**Acknowledgments:** The authors are grateful to D.A. Radchenko (Head of sector, Concern CSRI Elektropribor) for the support of manufacturing and development of GNSS/IMU gyrocompass prototype and model, I.Y. Vinokurov (First category engineer, Concern CSRI Elektropribor) for preparing and conducting the model car tests, P.N. Kostin (First category engineer) and P.Y. Petrov (Leading engineer, Concern CSRI Elektropribor) for the development of electronics and software for gyrocompass prototype.

**Conflicts of Interest:** The authors declare no conflict of interest.

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
