# Peer review of "Integrated GNSS/IMU-Gyrocompass with Rotating IMU. Development and Test Results"

_remotesensing, doi:10.3390/rs12223736_

Round 1
Reviewer 1 Report
Dear authors,
I have read your paper entitled: Integrated GNSS/IMU-Gyrocompass with Rotating IMU. Development and Test Results.
The paper is interesting, the advantage is, that it is based on real data, on the other hand, the methods can be compared to other one to see the performance of the used method.
I have several comments, you are using untraditional notation, the usage of K for azimuth, etc.
Why have you used 3 sigma, when in paper values with one sigma are presented?
You have mentioned, that GPS/GLONASS are used. Have you tested both systems together? In some data processing like RTK, it is not recommended to use both systems, because of not precise time synchronization.
Can you write me the type of gyroscopes and manufacturer?
Have you tested the performance using only GPS, only GLONASS, both of them?
Why have you used max. 5 satellites? You have mentioned, that PC and Matlab was used.
Have you analyzed the relation between number of satellites and final accuracy?
Thank you,
Reviewer
Author Response
Response to Reviewer 1 Comments
Dear colleagues, thank you very much for your review. Please, find our comments to your questions below.
Point 1:Why have you used 3 sigma, when in paper values with one sigma are presented?
Response 1: We are not quite sure that we understood your question correctly.
Probably, we are talking about the description of the research results [30-31], where the attitude errors at the 1 sigma level are given (p. 2 after figure.1). We have adjusted this level to 3 sigma and, accordingly, the level of errors. The text is corrected in lines 65-66: “3° (3σ)” instead of “1° (1σ)”, “0.3° (3σ)” instead of “0.1° (1σ)”.
Further, when analyzing the data, we considered only errors of the system at the 3-sigma level and maximum errors. Figures 10 (c) and 16 (c) with histograms are added to show heading error according to the tests path area where there is no special maneuvering and transients are ended (lines 390-392, 456-458).
Point 2:You have mentioned, that GPS/GLONASS are used. Have you tested both systems together?.In some data processing like RTK, it is not recommended to use both systems, because of not precise time synchronization.
Response 2: In the experiment, we did not process the signals from the GLONASS/GPS of both systems simultaneously. Indeed You are right there is a synchronization problem in case of using pseudorange and radial velocity while the attitude determiningin a tightly coupled INS. The authors are aware of this problem. However, it was not considered in the paper because in our case, when using phase measurements, the clock errors of different GNSS are almost completely eliminated, and therefore there are no obstacles to the simultaneous use of different GNSS signals.
Point 3:Can you write me the type of gyroscopes and manufacturer?
Response 3: Open loop FOG series ВГ910 are used. The manufacturer is CJSC Fizoptika, Russia. https://www.fizoptika.ru/catalog/gruppa-vg-910.
The text “open loop FOGs (CJSC Fizoptika, Russia)” and reference are added in line 128-129.
Point 4:Have you tested the performance using only GPS, only GLONASS, both of them?
Response 4: The results are presented separately when using GPS or GLONASS.Together they were not examined. However, considering to the achieved results, in our opinion, the attitude determination by the proposed method is possible with the simultaneous use of any existing GNSS signals.
Point 5:Why have you used max 5 satellites? You have mentioned, that PC and Matlab was used.
Response 5: There are two reasons why we used 5 satellites. Firstly, we took into account our experience of working with SNS compasses in a city environment, when a limited number of satellites are observed. Secondly, this is due to the limitations of the software developed at the time of the experiment.Indeed, an increase in the number of satellites used should reduce the heading error.
Point 6:Have you analyzed the relation between number of satellites and final accuracy?
Response 6: We investigated this issue in simulating and increased the satellites number up to 10. The research results show the more satellites the less heading error.It also corresponds to the experiment results while the satellites number increased from 1 to 5.
Thank your once again for the work done,
Sincerely yours

Reviewer 2 Report
Overall assessment
This article presents a complete study of a gyrocompass and its experimental validation which includes several significant innovations compared to current devices. The basic principle is to use a baseline consisting of two GNSS antennas and an Inertial Measurement Unit (IMU) both mounted on a platform rotating around an axis common to both sensors. The processing method based on an Extended Kalman Filter (EKF) combines an evolution model derived from navigation equations and a measurement model that uses acceleration and angular velocity measurements from the IMU and phase difference and pseudo-range and radial velocity measurements given by GNSS receivers connected to the antennas. The article focuses on determining the attitude angles of the carrier vehicle, in particular the heading.
The main innovation of the GNSS/IMU gyrocompass is based on the use of tactical-grade Fiber Optic Gyrometers (FOG) instead of Micro-machined ElectoMechanical System (MEMS) gyros, which allows the use of shorter length GNSS baseline with comparable performances in terms of heading accuracy (0,5 - 1° (3σ)). A second innovation consists of the use of a common reference clock for the two receivers dedicated to receiving signals from the GPS and GLONASS constellations, making it possible to avoid clock errors specific to each receiver. The design of this system and the processing method follow on from a great deal of prior research work recounted in the numerous bibliographical references in the article (41).
A prototype of the gyrocompas using two separate rotating platforms carrying respectively the GNSS antennas and the IMU was tested on board a road vehicle over a journey of almost 2 hours. An additional Inertial Navigation System was used conjointly in order to provide reference values of attitude angles. The results of this experiment are rather convincing. They notably show that the maximum heading error obtained with phase measurements coming from multiple GNSS satellites (1.2° with the vehicle moving) is comparable to that obtained with phase measurements coming from solely one GLONASS satellite (1.5° with the vehicle moving). The maximum heading error reaches 3° after one and a half hour when there is no GNSS signal available, and linear velocity measurements are used instead. This prototype opens up interesting prospects for the development of small gyrocompasses offering good performance whatever the configuration of the GNSS constellation or the quality of GNSS signal reception.
Overall, I found the article clear and detailed enough to lead the reader to understand the scientific approach used by the authors. My comments concern only a few points about the form of the text. The few rare typos found in the text (such as the lack of space after a comma) have been highlighted in the attached printed version.
Specific comments
Section 1: Introduction
This section is rather well done and gives a complete state of the art of GNSS/IMU based gyrocompasses both used on a rotating platform. In the first paragraph (lines 27-35), it seems that the instrument in question is a gyrocompass based only on GNSS and not GNSS/IMU. Even if as a reader I understand what "spaced antenna receivers" consist of, one additional sentence could be inserted (line 29) to explain that GNSS gyrocompass contains at least one baseline formed by two GNSS antennas. The remaining of the introduction is satisfactory.
Section 2: Main notations and reference frame definitions
No worries in this section: reference frames and mathematical notations used throughout the paper are clearly defined here.
Section 3: Materials
This section describes very well the composition of the GNSS/IMU gyrocompass. However, one detail of form bothers me: the great number of paragraphs with only one sentence (highlighted in yellow in the printed version of the paper). Authors would be well advised to group sentences together to form larger paragraphs in the text.
From line 133 to 138, the text includes statements on the interest of using rotating GNSS baseline which is not supported by any bibliographical reference, as done for IMU rotating platform (lines 139-143). It should be better to give at least one reference at this point.
Section 4: Methods
This section is really very clear and all the equations are given and their content well explained. I just highlighted the paragraphs that could be grouped together.
Section 5: Results ans discussion
The experience proposed in this section is quite appropriate for demonstrating the effectiveness of the GNSS/IMU gyrocompass and the data processing method used for heading determination. I particularly appreciated the fact that you represented the graphs of the heading error in the three calculation modes (phase measurements from multiple GNSS satellites, only one GNSS satellite and no phase measurement) over the entire time of the experiment. Thus it is possible to see the transient regime necessary for EKF stabilization, the effects of synchronisation errors between the gyrocompass and the reference INS and vehicle manoeuvres.
In addition to Figures 10(b), 13 and 16 showing the error of heading estimation, it would be interesting to represent the error histogram in a steady-state time interval between two vehicle manoeuvres. This would give an idea of the distribution of errors and would give an estimate of the accuracy of the device as a heading measuring system.

Author Response
Response to Reviewer 2 Comments
Dear colleagues, thank you very muchfor your review.
Please, find our comments to your questions below.
Point 1:Overall, I found the article clear and detailed enough to lead the reader to understand the scientific approach used by the authors. My comments concern only a few points about the form of the text. The few rare typos found in the text (such as the lack of space after a comma) have been highlighted in the attached printed version.
Response 1: Thank you very much for the good evaluation of our work. It is very valuable for us.In the references, missing space after a comma are added (lines 39, 66, 151).Corrected words: “notations” instead of “denotation” (lines 84, 91), “larger” instead of “large” (line 149). Changed the title of the section “Reference frame definitions and main notations” instead of “main denotations” (line 90)
Point 2:This section is rather well done and gives a complete state of the art of GNSS/IMU based gyrocompasses both used on a rotating platform. In the first paragraph (lines 27-35), it seems that the instrument in question is a gyrocompass based only on GNSS and not GNSS/IMU. Even if as a reader I understand what "spaced antenna receivers" consist of, one additional sentence could be inserted (line 29) to explain that GNSS gyrocompass contains at least one baseline formed by two GNSS antennas. The remaining of the introduction is satisfactory.
Response 2: Thank you very much again. The text is corrected: “Recently, devices contains at least one baseline formed by two global navigation satellite systems (GNSS) antennas have been widely used for attitude determination of mobile objects.” instead of ”Nowadays attitude determination is performed using technology based on measurement of carrier phase differences for the signals received by the spaced antennas from Global Navigation Satellite Systems (GNSS) receivers” (lines 27-31); “GNSS compasses” instead of “GNSS/IMU compasses” (lines 31, 33).
Point 3:This section describes very well the composition of the GNSS/IMU gyrocompass. However, one detail of form bothers me: the great number of paragraphs with only one sentence (highlighted in yellow in the printed version of the paper). Authors would be well advised to group sentences together to form larger paragraphs in the text.
Response 3: Larger paragraphs are formed (lines 124-130, 133-141).
Point 4:From line 133 to 138, the text includes statements on the interest of using rotating GNSS baseline which is not supported by any bibliographical reference, as done for IMU rotating platform (lines 139-143). It should be better to give at least one reference at this point.
Response 4: The reference is given (line 138).
Point 5:This section is really very clear and all the equations are given and their content well explained. I just highlighted the paragraphs that could be grouped together.
Response 5: Larger paragraphs are formed (lines 227-236, 250-259, 295-303, 335-338).
Point 6: In addition to Figures 10(b), 13 and 16 showing the error of heading estimation, it would be interesting to represent the error histogram in a steady-state time interval between two vehicle manoeuvres. This would give an idea of the distribution of errors and would give an estimate of the accuracy of the device as a heading measuring system.
Response 6: The error histograms are added in Figures 10 (c) and 16 (c) to show heading error according to the tests path area where there is no special maneuvering and transients are ended. Links to figures 10(c) and 16(c) are added in lines 390-391, 456-457. However, histograms appearance changes significantly while the vehicle maneuvering. The similar histogram for the Figure 13 is not included because of the low information content.
Thank your once again for the work done,
Sincerely yours
